# Scalable Production of Extracellular Vesicles and Its Therapeutic Values: A Review

**DOI:** 10.3390/ijms23147986

**Published:** 2022-07-20

**Authors:** Chiew Yong Ng, Li Ting Kee, Maimonah Eissa Al-Masawa, Qian Hui Lee, Thayaalini Subramaniam, David Kok, Min Hwei Ng, Jia Xian Law

**Affiliations:** 1Centre for Tissue Engineering and Regenerative Medicine, Faculty of Medicine, University Kebangsaan Malaysia Medical Centre, Jalan Yaacob Latif, Kuala Lumpur 56000, Malaysia; chiewyongng@gmail.com (C.Y.N.); liting1027@hotmail.com (L.T.K.); maimonah.almasawa@gmail.com (M.E.A.-M.); drygrasshui@gmail.com (Q.H.L.); subramaniamthayaalini@gmail.com (T.S.); davidkok.tl@gmail.com (D.K.); angela@ppukm.ukm.edu.my (M.H.N.); 2Faculty of Applied Sciences, UCSI University, Jalan Menara Gading No. 1, Kuala Lumpur 56000, Malaysia

**Keywords:** extracellular vesicle, large-scale, culture medium, production, stem cell, bioreactor, three-dimensional culture

## Abstract

Extracellular vesicles (EVs) are minute vesicles with lipid bilayer membranes. EVs are secreted by cells for intercellular communication. Recently, EVs have received much attention, as they are rich in biological components such as nucleic acids, lipids, and proteins that play essential roles in tissue regeneration and disease modification. In addition, EVs can be developed as vaccines against cancer and infectious diseases, as the vesicle membrane has an abundance of antigenic determinants and virulent factors. EVs for therapeutic applications are typically collected from conditioned media of cultured cells. However, the number of EVs secreted by the cells is limited. Thus, it is critical to devise new strategies for the large-scale production of EVs. Here, we discussed the strategies utilized by researchers for the scalable production of EVs. Techniques such as bioreactors, mechanical stimulation, electrical stimulation, thermal stimulation, magnetic field stimulation, topographic clue, hypoxia, serum deprivation, pH modification, exposure to small molecules, exposure to nanoparticles, increasing the intracellular calcium concentration, and genetic modification have been used to improve the secretion of EVs by cultured cells. In addition, nitrogen cavitation, porous membrane extrusion, and sonication have been utilized to prepare EV-mimetic nanovesicles that share many characteristics with naturally secreted EVs. Apart from inducing EV production, these upscaling interventions have also been reported to modify the EVs’ cargo and thus their functionality and therapeutic potential. In summary, it is imperative to identify a reliable upscaling technique that can produce large quantities of EVs consistently. Ideally, the produced EVs should also possess cargo with improved therapeutic potential.

## 1. Introduction

Extracellular vesicles (EVs) are naturally-occurring heterogeneous nano- to micro-sized lipid bilayer membrane vesicles packed with regulatory biological cargo, i.e., cytosol, lipids, proteins, and nucleic acids [1,2]. EVs are secreted by most of the cells and mediate intercellular communication in physiological and pathological conditions. EVs can be mainly categorized into three subtypes, i.e., exosomes, microvesicles (also called ectosomes), and apoptotic bodies, based on their biogenesis pathways. Each EV subtype has different sizes, cargoes, and functions [1]. Research on EVs as cell-free regenerative therapies, targeted therapies, drug carriers, diagnosis biomarkers, and cancer vaccines has grown drastically.

EVs have been widely investigated as drivers of tissue regeneration. In recent years, stem cell-derived EVs received much attention, as they possess therapeutic potential comparable to or even better than that of the parent cells [3]. In addition, stem cell EVs are nonliving vesicles with superior safety profiles to those of cells in clinical applications. Stem cell-derived EVs carry low risks of tumorigenicity and allogeneic immune rejection as well as minimal risk of microvascular occlusion during intravascular administration because of their nano size [4,5]. The regenerative potential of EVs has been reported in many preclinical studies to treat a myriad of diseases [6,7,8,9]. Additionally, positive outcomes have been reported in a few clinical trials, with many more EV-based clinical trials currently ongoing worldwide.

On the other hand, EVs of cancer cells have the potential to be used as drug carriers and cancer vaccines. Cancer cell-derived EVs have a higher affinity to cancer cells due to the unique protein and lipid composition that facilitates the binding or internalization of EVs in cancer cells [10,11,12]. Thus, they can be used for targeted delivery of chemotherapeutic agents and enhance treatment efficacy while minimizing off-target effects [13,14]. Cancer cell-derived EVs also possess large numbers of tumor antigens, which can trigger the host’s immune responses to inhibit tumor growth [15]. EVs produced by cancer cells can also be employed as diagnostic cancer biomarkers, since they contain cargo that reflects the tumor’s genetic and mutational status [16,17]. Understanding of the impact of tumor microenvironments, e.g., pH, extracellular matrix (ECM) stiffness, oxidative stress, hypoxia, and nutrient deprivation, as well as of treatment modalities, e.g., irradiation, chemotherapy, and photodynamic therapy, on the EV secretion of cancer cells has been instrumental in enhancing the EV production by cancer cells in vitro [18,19,20,21,22]. 

EVs can be derived from many sources, most commonly from cells and biofluids. Currently, EVs for preclinical and clinical studies are mainly produced from cultured cells, as they are easier to manipulate. However, the number of EVs produced by cells is meager in standard two-dimensional (2D) culture conditions. Thus, obtaining a sufficient EV yield is one of the major hurdles for translating EV-based therapeutics. Therefore, it is crucial to devise new strategies to stimulate cells to release more EVs for clinical applications. The manufacturing of EVs includes a series of sequential steps, beginning with the isolation and culture of source cells, which is followed by the separation and storage of EVs [23]. The current review focuses on evaluating strategies aimed at improving EV release, particularly that of small EVs, by cultured cells to meet the needs for large-scale production. Apoptotic bodies are not included in this review, as they do not fall within the scope of therapeutic agents.

## 2. Types of Extracellular Vesicles

EVs are broadly classified through different biogenesis pathways into three groups, i.e., exosomes, microvesicles, and apoptotic bodies (Figure 1). EV identification is generally performed through size and protein marker examination. Size evaluation is most commonly performed via nanoparticle tracking analysis (NTA), electron microscopy, dynamic light scattering (DLS), and tunable resistive pulse sensor (TRPS), while protein markers are detected through immunoblotting and immunosorbent assays [24]. In addition, flow cytometry has been valued to characterize the size, number, and purity of EVs robustly and reliably [25,26]. Nevertheless, it is still challenging to define and characterize the distinct subpopulations of EVs, since their sizes overlap and specific protein markers are lacking. Thus, EVs are widely divided by size into large and small EVs. Large EVs are those >200 nm in diameter, including oncosomes, apoptotic bodies, migrasomes, and microvesicles. On the other hand, small EVs are those <200 nm in diameter, such as exosomes and a subpopulation of microvesicles (ectosomes) [27]. In addition, exomeres—nonmembranous nanovesicles (<50 nm)—were also discovered to be secreted by various cells [28]. The cargo of the vesicles varies with their biogenesis, cell type, and physiological condition [29].

### 2.1. Exosomes

Exosomes are formed by an endosomal route. Exosomes are secreted by most cell types and can be detected in various body fluids. Exosomes play important biological roles in cell activities, cell–cell communication, cell maintenance, and immune response stimulation to maintain tissue homeostasis in normal physiological conditions. In addition, exosomes mediate the pathology of many diseases [1,30]. The cargo of exosomes varies according to the parent cells, as they may contain specific functional biomolecules. For example, B-lymphocytes and dendritic cells secrete exosomes expressing major histocompatibility complex (MHC) class II that activate T-cells and initiate immune response [31,32]. Furthermore, studies have shown that the same cell type can produce exosomes that are remarkably different in size and cargo. For example, polarized epithelial cells in the kidney tubule produce exosomes with distinct size and protein composition from the apical and basolateral plasma membranes [33,34].

Protein content in exosomes can be categorized based on the functions and sources of the proteins. Endosomal sorting complex required for transport (ESCRT) proteins are involved in exosome formation and multivesicular body (MVB) transportation, so ESCRT and associated proteins (e.g., Alix, TSG101, HSC70, and HSP90β) have been found in exosomes secreted by all cell types [1]. Cytoskeletal proteins (e.g., actin, syntenin, and moesin), signal transduction proteins (e.g., kinase proteins), and metabolic enzymes (e.g., GAPDH, LDHA, PGK1, aldolase, and PKM) with cell support function have also been detected in exosomes [35]. Exosomes are enriched in glycoproteins compared with their parent cells and have low levels of proteins associated with the Golgi apparatus and endoplasmic reticulum [1]. The enriched lipid content, i.e., cholesterol, ceramide, sphingomyelin, and glycerophospholipids, is mostly localized in the exosomal membrane [36]. The nucleic acid contents of exosomes with known functions are mRNAs, miRNAs, and siRNAs [37].

### 2.2. Microvesicles

Microvesicles are another group of EVs that involve intercellular communication besides exosomes. Microvesicles are formed through direct outward budding of the cell’s plasma membrane. The proteins specifically associated with organelles such as the mitochondria, Golgi apparatus, nucleus, and endoplasmic reticulum have been predicted to be depleted in microvesicles, as these organelles are not related to its biogenesis. Nonetheless, microvesicles were shown to carry mitochondria [38]. Microvesicles contain mainly cytosolic and plasma membrane-associated proteins, especially tetraspanins, which are rich in the plasma membrane. Other enriched proteins include cytoskeletal proteins, heat shock proteins (HSPs), integrins, and proteins with posttranslational modifications, including glycosylation and phosphorylation [1]. Similarly to exosomes, microvesicles contain DNAs, RNAs, proteins, and bioactive lipids that can be transferred to acceptor cells in the cell–cell communication process [39,40,41,42]. The pleiotropic effects of microvesicles have gained more interest in recent years [43,44].

### 2.3. Apoptotic Bodies

Apoptotic bodies are a subtype of apoptotic EVs (ApoEVs) secreted by apoptotic cells [45]. They are products of programmed cell death, formed after apoptotic cells undergo the blebbing and fragmentation of the plasma membrane. The composition of apoptotic bodies is very distinct from that of exosomes and microvesicles, as they might contain cellular components such as intact organelles, chromatin, fragmented DNA, micronuclei, cytoplasm content, and degraded proteins [46]. The protein profile of apoptotic bodies is similar to that of the cell lysate, i.e., it has high levels of proteins associated with the nucleus (e.g., histones), mitochondria (e.g., HSP60), Golgi apparatus, and endoplasmic reticulum (e.g., GRP78) during analysis [1]. Apoptotic bodies can transport biomolecules, including miRNAs and DNAs, to modulate intercellular communication [47,48]. Defective release and clearance of these vesicles have been linked to various autoimmunity-related conditions [49]. A recent study showed that efferocytosis of apoptotic vesicles contributed to macrophage hemostasis and type II diabetes therapy, which highlighted the functional capability of apoptotic bodies [50].

The operational terms for EVs are used in this review because the biogenesis of the collected EVs cannot be determined and because of the lack of specific EV markers to determine the subcellular origin of each subtype. Here, EVs are classified based on their sizes and biochemical compositions.

## 3. Therapeutic Value of Extracellular Vesicles

Because of the lipid bilayer membrane, the therapeutic cargos of EVs are protected from degradation by enzymes in the biofluids. They thus can deliver their bioactive cargoes from parental cells to acceptor cells over a distance [51]. Since EVs can target specific recipient cells and a range of biomolecular cascades, they are considered promising therapeutic agents in regenerative medicine. A recent review suggested that EVs derived from a variety of cell types, including MSCs, epithelial cells, endothelial progenitor cells, dendritic cells, macrophages, and T cells, could modulate intracellular signaling pathways related to cell viability, ECM interaction, angiogenesis, and immune responses that are crucial in regenerative medicine [52]. EVs were used to promote the regeneration of many tissues and organs, and their regenerative potential was on par with that of cell therapy [53]. EV-based therapy mitigated some concerns raised by cell-based therapy with features such as a low risk of malignancy, the capability of crossing the biological barrier, and a low immunogenic profile [54]. Furthermore, EVs are versatile and can be combined with scaffolds to enhance their regenerative potential [52]. The therapeutic efficacy of EVs can be further enhanced by modifying their cargoes with specific therapeutic agents, including drugs, chemotherapy, proteins, lipids, nucleic acids (such as mRNA, miRNAs, siRNAs, and snoRNAs), and nanoparticles [55,56].

EVs also appear as therapeutic targets. Diseased cell-derived EVs carry pathological cargoes, which are transported to the normal cells that up taken the EVs, thus affecting the biological functions of the normal cells [27,57]. It has been shown that small EVs released from senescence cells transfer prosenescence signals to younger cells and stimulate them to acquire senescent phenotypes [58]. In the tumor microenvironment, EVs play a crucial role in tumor progression by transferring oncogenic entities to cancer or noncancer cells [59]. Tumor cell-derived EVs promote angiogenesis and vascular permeability and suppress immunity toward tumor cells, which activities drive tumor growth and malignancy [60,61]. Moreover, tumor-derived EVs induce cellular transformation of noncancer cells [62,63]. Inhibiting exosome secretion also suppresses the directional persistence and speed of the migration of tumor cells [64]. Hence, EV-targeting therapy has been proposed as a new approach to inhibiting disease progression by eliminating circulating disease-associated EVs, reducing disease-associated EV secretion, and disrupting the uptake of disease-associated EVs [65].

EVs have emerged as informative biomarkers with the potential to become a valuable tool for the diagnosis and treatment monitoring of various diseases. EVs can be used as biological indicators for cardiometabolic diseases [66,67,68,69], neurological diseases [70,71,72,73], liver diseases [74,75], kidney diseases [76], respiratory diseases [77], skin diseases [2,78], and detection of graft rejection [79]. Small EVs from oral biofluids (i.e., saliva and gingival crevicular fluid) may act as biomarkers for diagnosing oral diseases noninvasively [80,81]. Additionally, a recent study identified detectable changes in salivary exosome proteins and miRNAs before and after work shifts, showing their potential as biomarkers for cognitive fatigue [82]. In addition, EVs also demonstrated the capability as biomarkers to diagnose cancer, monitor cancer progression, detect cancer recurrence, and examine therapeutic response [65,83,84]. The main advantage of EVs as biomarkers is they can be collected from almost all biofluids, such as urine, oral biofluids, blood, breast milk, and cerebrospinal fluid (CSF), via noninvasive or minimally invasive techniques, which reduces patient inconvenience, increases the speed of analysis, and decreases analytical costs [85]. The complexity of EV subgroups is influenced by the donor cell type, the condition of cellular activation, the local microenvironment, the biogenesis mechanism, and the intracellular cargo-sorting pathway, contributing to a significant variation in EV profiles between patients and healthy controls [86]. Because of these advantages, the use of EVs as biomarkers permits longitudinal patient sampling, supports clinical decision-making in early diagnosis and prognosis, and allows therapy monitoring in various diseases.

EVs are ideal candidates as carriers for drug delivery because of their ability to carry cargo between cells [86,87,88,89]. The hydrophobic molecules are preferentially intercalated to the bilayer membrane, while the hydrophilic molecules are concentrated at the lumen. Traditional drug delivery systems demonstrated a major drawback, i.e., failure to deliver macromolecules and nanoparticles across biological barriers to reach the target tissues or intracellular targets [90]. Furthermore, there are concerns about the immunogenicity and toxicity of nonnatural delivery mechanisms. EVs, on the other hand, appear to possess many of the characteristics of a promising carrier system. They are natural nano-sized vesicles that can pass through biological barriers, are readily uptaken by cells, and have low immunogenicity. Furthermore, as mentioned above, the bilayer membrane can protect the contents of EVs from degradation and stabilize the EVs in biofluids [88]. Furthermore, exosomal proteins such as CD47 protect exosomes from phagocytic clearance by monocytes, thus extending half-life in circulation compared with that of liposomes [91]. EV uptake by cells is not a random process but is highly dependent on the specific surface receptor and ligand interactions between the cells and EVs [92]. The specific delivery of EVs to target cells can be further enhanced through surface protein modification [93]. Alternatively, EVs can be loaded with magnetic nanoparticles and localized at the target site using an external magnetic field [94].

EVs also have the promise to be cell-free vaccines [95,96]. A vaccine is a biological product containing antigens that are used to stimulate the body’s immune response and provide protection against infection and/or disease [97]. EVs from mammalian cells and bacteria carry a variety of cellular components, such as antigenic determinants and virulence factors, that can lead to immunomodulatory effects [98]. These antigenic determinants and virulence factors can also be loaded into EVs exogenously, allowing the EVs to function as antigen presenting. Thus, EVs can be used as vaccine carriers. Currently, researchers are focusing on utilizing exosomes produced by cancer cells as cancer vaccines and utilizing exosomes derived from microorganisms, as well as engineered cells, as vaccines to protect the host against infectious diseases [99,100,101]. The EV-based vaccine against cancer has garnered the most attention for preventing tumor development or treating existing tumors [102]. Compared with conventional vaccines, EV-based vaccines offer unique traits valuable in vaccine design, such as better biosafety and efficiency as antigen-presenting systems and adjuvants [103]. Furthermore, cell-free EV-based vaccines offer several advantages over traditional cell-based vaccines, including increased stability and ease of storage for long periods without significant loss of activity [98].

Nonetheless, several hurdles to clinical applications of EVs as cell-free therapy, such as the short half-life in circulation of EVs; the lack of well-designed clinical trials; and the lack of appropriate dosages, administration methods, and timing, have yet to be addressed [54]. Though the mechanism underlying EV-induced cancer initiation and progression, as well as those of other diseases, has been partially uncovered, the use of EVs as therapeutic targets, biomarkers, and cell-free vaccines is not yet mature because of the high heterogeneity of EVs and the possibility that different EV subtypes may perform differently [104]. Furthermore, loading EVs with assorted cargo and pharmaceutical agents is not easy and needs to be optimized. Hence, further research on EV therapeutic application must be done, and EV upscaling is one of the directions this research must take.

## 4. Strategies to Increase Production of Extracellular Vesicles

Several strategies have been investigated in the literature to upscale the release of EVs from cultured cells. Alteration of cell culture environments, e.g., three-dimensional (3D) culture, chemical stimulation, physical stimulation, physiological modification, and genetic manipulation of source cells, has been the most common approach to enhancing the quantity and quality of EVs secreted by cells. Alternatively, some studies have explored the use of physical techniques such as sonication, nitrogen cavitation, and porous membrane extrusion to produce EV-like vesicles, known as EV-mimetic nanovesicles, which have similar characteristics and functionalities to those of natural EVs while circumventing some of their limitations (Figure 2). Table 1 shows the strategies used to increase EV production.

### 4.1. Three-Dimensional Culture

One technique for increasing EV secretion from cultured cells is using 3D cell culture systems such as bioreactors and cell spheres. Generally, 3D cell culture systems allow the expansion of a large number of cells in high density for high yields of EVs, as they provide a large surface area for cell growth [169].

The bioreactor is the most commonly used 3D cell culture system for large-scale EV production, as not only does it increase the volume of production, but the mechanical stimulus provided by the bioreactor stimulates the cultured cells to produce more EVs [170]. Generally, the use of bioreactors can increase the yield and concentration of EVs in conditioned medium and reduce the production time. In a study, the authors found that umbilical cord-derived MSCs (UC-MSCs) cultured in 3D hollow fiber bioreactors secreted 7.5 times more small EVs than cells cultured in 2D tissue culture flasks [105]. In addition, the authors found that 3D-EVs were more potent in promoting chondrocyte proliferation and migration as well as inhibiting chondrocyte apoptosis in vitro. Using an in vivo cartilage defect model, the authors proved that 3D-EVs were more effective in promoting cartilage regeneration. It was postulated that 3D-EVs modulated chondrocyte functions by activating the TGF-β1 and Smad 2/3 signaling pathways.

Cao et al. demonstrated that the protein yield of small EVs from UC-MSCs cultured in a 3D hollow-fiber bioreactor was 19.4-fold higher than that from cells cultured in a 2D conditions [106]. In vivo findings showed that 3D-EVs were more effective than 2D-EVs in ameliorating cisplatin-induced acute kidney injury, as indicated by better renal function, less severe pathological changes in renal tubules, and lower infiltration of inflammatory cells. The authors attributed the renoprotective effects of 3D-EVs to their increased uptake by tubular epithelial cells and enhanced antiinflammatory activity.

In a study comparing the small EVs secreted by MSCs derived from bone marrow (BM), adipose tissue (AT), and the umbilical cord matrix (UCM) cultured in a 3D microcarrier-based Vertical-Wheel™ Bioreactor (VWBR) and a 2D tissue culture flask, the authors found 4.0-fold, 4.4-fold, and 8.8-fold increases in small EV concentration in conditioned medium and 1.4-fold, 3.7-fold, and 3.9-fold increases in small EV productivity for the 3D cultured BM-MSCs, AT-MSCs, and UCM-MSCs, respectively, compared with the 2D cultures [107]. The therapeutic potential of the isolated EVs is unknown, as no functionality assessments were performed in this study.

Watson et al. reported a 40-fold increase in small EV secretion when human embryonic kidney (HEK) 293 cells stably expressing hetIL-15 were cultured in a hollow fiber bioreactor than when they were cultured in a standard 2D culture [108]. Moreover, the study found that the 3D-EVs contained fewer serum protein contaminants than the 2D-EVs. Large-scale production of EVs using bioreactors has also been reported in other studies [109,171]. However, these studies did not compare the EV yield with that of 2D cultures. Nonetheless, data from these studies demonstrated the feasibility of large-scale production of cell-derived EVs using bioreactors.

In addition, MSCs cultured in collagen scaffolds were found to secrete two times more small EVs than those cultured in 2D conventional conditions [110]. The 3D-EVs were more effective in promoting neurological functional recovery of traumatic brain injury models in rats than 2D-EVs and liposomes. The 3D scaffold is often used with a bioreactor, which helps to improve the nutrient perfusion and waste removal, which is critical to keep the cells viable and healthy. Patel et al. cultured human dermal microvascular endothelial cells (HDMECs) in a 3D-printed scaffold-perfusion bioreactor to collect small EVs [111]. The authors found that 3D-cultured endothelial cells secreted 100 times (collected on day 1) and 10,000 times (collected on day 3) more small EVs than those cultured in static scaffold and tissue culture flasks as assessed by NTA. However, a mere 14-fold increase in CD63 Exo-ELISA analysis was recorded between the 3D-EVs and the 2D-EVs collected from the tissue culture flasks. Furthermore, the 3D-EVs preconditioned with ethanol demonstrated a more potent provascularization effect that was attributed to higher concentrations of the proangiogenic lncRNAs HOTAIR and MALAT1 in 3D-EVs. 

Apart from bioreactors, 3D spheroid cultures have been found to increase the secretion of EVs by cells [112]. 3D hanging drop spheroids and 3D poly(2-hydroxyethyl methacrylate) spheroids produced significantly more EVs than 2D-cultured BM-MSCs [112]. Interestingly, the authors found that EV secretion reduced when the size of the 3D spheroids increased. Thus, size could be an important parameter to optimize when 3D spheroid culture is used for the large-scale production of EVs.

### 4.2. Physical Stimulation

Physical stimulation techniques such as irradiation, electrical stimulation, magnetic field stimulation, mechanical stimulation, and topographic cues have been explored and used to enhance EV production. Generally, physical stimulation stresses cells to produce more EVs [172]. Ionizing radiation has been reported to increase the number of EVs produced by cancer cells in a dose- and time-dependent manner [18,113,114,115]. The upregulation of EV production by cancer cells upon exposure to ionizing radiation has been linked with the DNA-damaged activated p53 signaling pathway [173,174]. These studies indicated that irradiation could increase the release of EVs from cancer cells, thus aiding in developing strategies for cancer treatment. However, a study found that ionizing radiation neither altered EV secretion by cancer cells nor modified the protein cargo of the secreted EVs [175]. Limited research has explored the effects of ionizing radiation on EV secretion by normal cells. A study found that ionizing radiation increased the small EV particle concentration of astrocytes by 1.71 times [115]. In addition, nonionizing radiation (ultraviolet radiation) [116], photodynamic therapy (Foscan^®^ photosensitizer) [19], and acoustic irradiation (at low power and high frequency) [117] have been reported to increase the number of EVs secreted by cancer cells. 

Low-level electrical stimulation applied to murine melanoma and fibroblast cells was found to stimulate EV secretion, possibly through Rho guanosine triphosphatase (GTPase) activation [118]. In another study, focal and transient electrical stimulation that induced cell membrane nanoporation increased the EV secretion of mouse embryonic fibroblasts 50-fold, while moderate increases in EV yield were detected in cells cultured in serum-depletion conditions, hypoxic conditions (1% O_2_) and heat stress conditions (42 °C for 2 h) [119]. Additionally, nanoporation increased the mRNA cargo of EVs. The combination of magnetic iron (III) oxide nanoparticles (Fe_3_O_4_) and a static magnetic field (SMF) was utilized to increase the small EV secretion of bone MSCs [120]. The highest EV production was recorded in the Fe_3_O_4_ + SMF group, followed by the Fe_3_O_4_ and untreated groups. In terms of functionality, the small EVs secreted by the Fe_3_O_4_ + SMF-stimulated cells showed better osteogenic and angiogenic potential than the Fe_3_O_4_ stimulated cells and naïve cells.

Several studies have reported higher EV production by cancer cells cultured on biomaterials with higher stiffness; increased ECM stiffness is one of the key changes in the tumor microenvironment [20,176]. EVs induced by stiff matrices were found to promote tumor cell migration and proliferation. Cyclic stretch increased the CD63+ EV secretion of periodontal ligament cells, and the secreted EVs demonstrated improved immunomodulatory properties to suppress IL-1β production by activated macrophages [121]. Zhang et al. cultured BM-MSCs on micro-/nanonet-textured hierarchical titanium surfaces and micro-/nanotube-textured hierarchical titanium surfaces and found that these surface topographies increased small EV secretion [122]. 

Multiple studies have tested nanoparticle incorporation to increase the number of EVs secreted by cultured cells. The nanoparticles found to increase EV secretion include platinum nanoparticles in human lung epithelial adenocarcinoma cancer cells (through induction of oxidative stress and ceramide pathway) [123], silver-titanium oxide nanoparticles in B16F1 mouse melanoma cells (through induction of oxidative stress) [146], poly(lactic-co-glycolic acid)-polyethyleneimine (PLGA-PEI) positively charged-surface-modified nanoparticles containing iron oxide in MSCs (through promotion of MVB formation) [177], and calcium phosphate particles in macrophage-like RAW264.7 cells and monocyte-like THP-1 cells (through promotion of MVB formation and fusion with the plasma membrane) [133]. In addition, bioactive glass upregulated the small EV production of MSCs through the activation of the nSMase and Rab GTPase pathways [124].

### 4.3. Chemical Stimulation

The addition of chemicals to the culture medium to boost EV secretion has been investigated in several studies. Treatment with sodium iodoacetate (IAA; glycolysis inhibitor) and 2,4-dinitrophenol (DNP; oxidative phosphorylation inhibitor) was found to increase the number of small EVs secreted by cancer cells [125]. In addition, IAA/DNP increased the number of small EVs released into the culture medium from kidney explants. In vivo validation of the in vitro and ex vivo data was achieved by injecting the IAA/DNP into mice. In comparison with that in control mice, IAA/DNP injection increased the quantity of small EVs in the blood. The study partially related the higher small EV production to higher intracellular 2′-3′-cAMP levels. Wang et al. tested the effects of several small molecules, including fenoterol, norepinephrine, N-methyldopamine, mephenesin, and forskolin, on the small EV production of BM-MSCs [126]. The findings showed that all of these small molecules could stimulate small EV secretion and that the magnitudes of the increases were affected by the concentrations of each small molecule. Synergistic improvement in exosome secretion was recorded for the combinations of norepinephrine with forskolin and norepinephrine with N-methyldopamine, but not for the combination of N-methyldopamine with forskolin. Furthermore, a multiple-component herbal combination in the Suxiao Jiuxin pill (a traditional Chinese herbal medicine) revealed a synergistic effect in promoting the small EV secretion of cardiac MSCs via a GTPase-dependent pathway [127].

In addition, adiponectin, an adipokine, increased the numbers of small EVs produced by MSCs and vascular endothelial cells through T-cadherin [128,129]. In addition, adiponectin increased the concentration of exosomes in mouse serum, and this increase was linked to the augmented cardioprotective function of primed MSCs. Docosahexaenoic acid (DHA) increased the CD63+ EV secretion of breast cancer cells. The DHA-EVs contained more antiangiogenic miRNAs (miR23b, miR-27b, and miR-320b), which aided in anticancer action [130].

Intracellular calcium ions were reported to modulate EV release [131,132,178]. A few studies explored the effects of calcium exposure on EV production [133,134]. The results indicated that calcium exposure increased EV production. In a different study, extracellular DNA and phosphorothioate CpG oligodeoxynucleotides were found to induce Alix+ EV secretion of HEK293 cells and head kidney leukocytes of Atlantic salmon [135]. The neutral and cationic liposomes were reported to stimulate the EV secretion of tumor cells in a dose-dependent manner [136]. However, the PEGylated liposomes diminished the EV production. Thus, the authors postulated that the influence of liposomes on EV production was dependent on their physicochemical properties.

### 4.4. Physiological Modification

The quantity and functionality of cell-secreted EVs are likely microenvironment dependent. In the physiological state, the oxygen level in peripheral tissues, known as “physoxia”, ranges between 1 and 11% [179]. Multiple studies have reported that hypoxic preconditioning augments the therapeutic efficacy of MSCs [180,181,182]. The improved therapeutic efficacy could be related to the higher EV secretion. Dong et al. showed that UC-MSCs cultured in hypoxic conditions (5% O_2_) demonstrated a higher proliferation rate and viability than those cultured in normoxia (21% O_2_) [138]. The hypoxic UC-MSCs produced more small EVs with higher potency to attenuate chronic airway inflammation and lung remodeling in ovalbumin-induced asthma mice. The results of another research group showed that hypoxic BM-MSCs secreted more small EVs with more potent therapeutic potential in promoting cartilage [139] and spinal cord regeneration [140] than those secreted by normoxic cells. Secretion of CD29+, CD44+, CD73+, CD31−, and CD45− EVs from UC-MSCs also increased in hypoxic conditions, and the angiogenic potential of the secreted EVs was superior for endothelial cells that were cultured in hypoxic conditions than those that were cultured in normoxic conditions [141]. In addition, extreme hypoxic conditions (0.5% O_2_) remarkably increased small EV release by MSCs [142]. In terms of functionality, hypoxic small EVs were more effective in promoting myocardial repair than normoxic small EVs, as they promoted vascularization, reduced cardiomyocyte apoptosis, reduced scar tissue formation, and enhanced the recruitment of cardiac progenitor cells. However, contradictory results were reported by Almeria et al., who found no significant difference in EV secretion by AT-MSCs cultured in normoxic and hypoxic conditions [183]. However, hypoxic priming enhanced the angiogenic potential of the secreted EVs. Apart from stem cells, hypoxic conditioning also has been found to induce EV secretion by cancer cells [22,114,116,137,184].

Since hypoxic conditioning can promote EV production, Gonzalez-King et al. overexpressed hypoxia-inducible factor-1a (HIF-1a), a vital mediator in low oxygen adaptation, in human dental pulp MSC via lentiviral transduction [143]. The HIF-1a overexpressed MSCs produced more small EVs that showed more potent angiogenic potential. Taken together, hypoxia culture appears to be more suitable for maintaining both stem cell and cancer cell cultures, as it mimics the native tissue physiological microenvironment. Stem cells and cancer cells cultured in hypoxic environments are also more biologically active, secreting more EVs.

Anderson et al. primed BM-MSCs under peripheral arterial disease (PAD)-like conditions, i.e., 0% serum and 1% oxygen [144]. They found that BM-MSCs cultured in PAD-like conditions secreted more low-density EVs and fewer high-density EVs than the control cells. Furthermore, the exosomes derived from PAD-like culture elevated the expression of several proangiogenic signaling proteins. This study showed that serum and oxygen deprivation could be used in tandem to induce low-density EV secretion. The use of serum deprivation to induce the EV secretion of mouse embryonic fibroblasts was reported by Yang et al. [119].

Enhancement of EV production via thermal stimulation was reported by Hedlund et al. [21]. Induction with thermal (40 °C for 1 h) and oxidative stress (50–100 µM H_2_O_2_ for 2 h) increased the secretion of CD63+ EVs by leukemia/lymphoma T- and B-cells [21]. Interestingly, the findings also showed that cells responded to stressors differently. The Jurkat cells were more responsive to oxidative stress, and the Raji cells were more susceptible to thermal stress by producing more CD63+ EVs in these culture conditions. Thermal stimulation of EV production by B-cells was also reported by Clayton et al., who found a small (1.25-fold) increase in EV secretion when cells were cultured at 40 °C for 3 h [145]. Harmati et al. showed that B16F1 mouse melanoma cells produced more small EVs when they were cultured at 42 °C for 2 h three times [146], and Gong et al. found that MGC-803 human gastric cancer cells released more small EVs in response to high temperatures (40 °C) [116].

Physiological and intracellular pH are important in many biological processes and cellular metabolism [185,186]. Kim et al. reported that a slight difference in the pH of the culture medium affected cell reprogramming and differentiation [187]. Besides hypoxia, extracellular acidity is another hallmark of cancer because of the accumulation of glycolytic metabolites such as lactic acid [188]. Melanoma cells secreted more Lamp-2+, CD81+, and Rab5B+ EVs that served as intercellular cross-talk mediators in acidic conditions (pH 6.0) than in buffered conditions (pH 7.4) to transport tumor-associated proteins to the other cells [12]. Higher secretion of small EVs in acidic pH (pH 4) was also reported by Gong et al. using MGC-803 human gastric cancer cells [116]. In a study using HEK293 cells, the authors reported that more CD9+, CD63+, and Hsp70+ EVs were collected from a conditioned medium of HEK293 cells cultured at pH 4 than from conditioned media of cells cultured at pH 7 and pH 11 [147]. The findings above demonstrated that cells could respond to environmental stresses and pathological conditions, such as tumor microenvironment (i.e., higher temperature, lower oxygen tension, and lower pH level), by altering their EV production. The changes in EVs produced by cells are also being studied to understand the cellular response to stress and pathological conditions.

### 4.5. Genetic Manipulation

Genetic modification of parental cells has been performed to modulate the signaling pathway regulating EV secretion. Rab proteins are GTPases that regulate vesicle traffic and have been identified to regulate the secretory pathway of EVs [189]. Studies by Bobrie et al. [190] and Ostrowski et al. [150] showed that silencing of *RAB27A* and *RAB27B* genes reduced multivesicular endosome (MVE) docking to the plasma membrane and reduced the small EV secretion of cancer cells. Furthermore, Rab35 depletion resulted in the accumulation of late endosomal vesicles and reduced exosome secretion in the oligodendroglial precursor cells [191]. More recently, Rab13 and Rab7a were also found to regulate the EV secretion of cancer cells [149,192]. 

Phospholipase D (PLD) catalyzes phosphatidylcholine hydrolysis to produce phosphatidic acid, an important lipid messenger involved in cell signaling, including exocytosis and endocytosis [193,194]. A significant increase in BODIPY-ceramide-labeled EV secretion was reported in PLD2-overexpressing RBL-2H3 cells (mast cells), while PLD2-knockout RBL-2H3 cells demonstrated poorer EV release [148]. PLD2 has been found to act as the effector of the ADP ribosylation factor 6 (*ARF6*) gene in regulating intraluminal vesicle (ILV) budding, thus playing an integral role in exosome production [195]. Furthermore, ARF6 has been reported to control the shedding of microvesicles in tumor cells [196]. Böker et al. found that the increased exogenous expression of tetraspanin CD9 after lentivirus transduction enhanced the secretion of small EVs in five cell lines, i.e., HEK293, SH-SY5Y, HeLa, Raji, and Jurkat, up to threefold [151]. Apart from the molecular pathways reported above, modulation of P2X7 and SNAREs receptor expression was explored to upregulate EV secretion, as these proteins have been found to influence EV formation, trafficking, and secretion in cells [197,198]. The results from these studies indicated that EV production could be modulated by targeting the key factors involved in the biogenesis and release of EVs through genetic modification of parent cells.

As discussed above, stress modulates the production of EVs by cells [146,199]. At the molecular level, p53-regulated exosome production typically occurs in response to the stress from DNA damage. Yu et al. demonstrated that upregulated transcription of tumor suppressor-activated pathway 6 (TSAP6) by activated p53 upon γ-radiation increased the small EV secretion of H460, a non-small cell lung cancer cell line [200]. It was also found that transfection of HA-tagged TSAP6 into H460 cells allowed small EV secretion without stress stimuli. Impaired EV secretion in TSAP6-knockout mice was reported by Lespagnol et al. [201]. These results indicated that TSAP6 was an essential mediator of p53-regulated exosome production. In addition, transfection of liver kinase B1 (LKB1), which is known to modulate the cell functions through the p53 pathway in lung cancer cells, was also found to increase the secretion of small EVs [152]. The small EVs secreted by LKB1-expressing cells contained fewer migration-suppressing miRNAs that inhibit cell migration.

Upregulation of eukaryotic translation initiation factor 3 subunit C (EIF3C) in human hepatocellular carcinoma cells increased the secretion of proangiogenic small EVs [153]. Knockdown of PIKfyve increased the small EV secretion of human prostate cancer epithelial cells by inhibiting MVB and autophagosome fusion with lysosomes and increasing the fusion of MVBs and autophagosomes [154].

Cell immortalization is a technique utilized to achieve the consistent production of EVs on a large scale using the desired cell source. Chen et al. generated highly expansible human ESC-MSCs by transfecting cells with the *MYC* gene [155]. The immortalized *MYC*-transformed hESC-MSCs bypassed cell senescence and could maintain high proliferation for more than 20 passages. Notably, the small EVs secreted by the immortalized *MYC*-transformed hESC-MSCs exhibited cardioprotective potential in an in vivo myocardial ischemia/reperfusion injury model. In another study, the same group of researchers revealed the safety of daily injection of the immortalized *MYC*-transformed hESC-MSC-secreted small EVs, as the small EVs did not affect tumor growth [202]. Nonetheless, the effects of EVs on cancer progression warrant further examination, as mixed results have been reported in different studies [203,204,205,206].

### 4.6. Preparation of EV-Mimetic Nanovesicles

Apart from stimulating the cultured cells to produce more EVs, an engineered approach has been developed to produce large-scale mimetic biologically functional nanovesicles, known as EV-mimetic nanovesicles. EV-mimetic nanovesicles are synthetic EVs that can be produced via top-down (plasma membrane fragmentation) or bottom-up (supramolecular chemistry) techniques [207]. EV-mimetic nanovesicles possess properties like those of naturally secreted EVs in terms of morphology, size, and functions [208]. Physical techniques, such as nitrogen cavitation, porous membrane extrusion, and sonication, and chemical techniques using chemical agents have been utilized to disrupt the cellular membranes and then the self-reassembly of lipids and membranes to form lipid vesicles that contain active ingredients. The advantages of this method are that it is easy to perform, increases the EV yield, and permits the production of homogenous EVs on a large scale.

Cavitation is a technical word in physics that describes the creation of many microscopic vapor-filled cavities as a result of a rapid pressure change in a liquid. When these cavities collapse, they produce a powerful shock that causes items to shatter. Nitrogen cavitation refers to the use of nitrogen gas to provide the pressure necessary for cavitation forces to occur. Gao et al. were the first to report the use of nitrogen cavitation to create synthetic EVs from white blood cells [156,209]. The cells were broken by expanding bubbles, which released cellular components into the fluid. Broken cellular membranes created vesicles with a wide range of particle sizes on their own. Gao et al. discovered that 50–75% of the cell plasma membrane generated vesicles with diameters of 180–200 nm. Extrusion of the vesicles through a membrane with 200 nm pore size resulted in EV-mimetic nanovesicles of uniform size. Nitrogen cavitation produced 16 times more EV-mimetic nanovesicles than naturally secreted EVs.

Sonication is regularly used for liposome preparation [210]. However, it can also be employed for EV-mimetic nanovesicle preparation. Thamphiwatana et al. used sonication to prepare EVs from macrophages [157]. In their protocol, the membranes of mouse macrophages were purified using a combination of hypotonic lysis, mechanical disruption, and differential centrifugation before sonication to form membrane vesicles that were later fused onto a poly(lactic-co-glycolic acid) (PLGA) core. The EV-mimetic nanovesicles retained many of the biological properties of the macrophages and were able to treat sepsis in a mouse bacteremia model. In another study, the authors prepared EV-mimetic nanovesicles through sonication of human UC-MSCs [158]. The EV-mimetic nanovesicle yield from sonication preparation was approximately 18.5-fold higher than the yield of natural EVs secreted by cells cultured in a serum-depleted medium. The EV-mimetic nanovesicles prepared by sonication were slightly larger than the naturally secreted EVs, i.e., 133.3 ± 1.8 nm vs. 122.9 ± 2.3 nm, respectively. Nonetheless, both expressed EV markers, i.e., CD9, CD63, and CD81. In terms of functionality, both EV preparations could promote wound healing in vivo using a mouse full-thickness excisional wound model.

A few studies have used the serial extrusion technique to prepare EV-mimetic nanovesicles. In general, in this technique, cells are mechanically broken down into nanosized vesicles by passing them through filters with reducing pore sizes (e.g., 10, 5, and 1 µm). The produced EV-mimetic nanovesicles can be up to 100 times more abundant than naturally secreted EVs, and they share the common features of exosomes, including size (30–200 nm) and marker expression (e.g., positive for CD9, CD63, CD81, TSG101, moesin, and β-actin) [159,160,211,212]. EV-mimetic nanovesicles could be used as drug carriers [159,160,212]. They were also more effective than naturally secreted EVs in treating emphysema [161] and could induce liver regeneration [162] in vivo.

Chemical agents, such as alkaline solutions, can be used to break down the cell membrane. Under sonication, the membrane components may self-assemble to form EV-mimetic nanovesicles after neutralizing the pH. Go et al. used human U937 monocytes to make EV-mimetic nanovesicles via sequential treatment with alkaline solution and sonication with and without the presence of dexamethasone [163]. The EV-mimetic nanovesicles exhibited identical physical properties to spontaneously released EVs. In comparison with the cell culture approach, there was a 200-fold increase in EV generation. The authors also discovered that the EV-mimetic nanovesicles lacked intracellular compartments such as cytosolic proteins and nucleic acids. In terms of functionality, the EV-mimetic nanovesicles loaded with dexamethasone were able to reduce systemic inflammatory response syndrome (SIRS) caused by the outer membrane vesicles (OMVs) of Gram-negative bacteria.

Sulfhydryl-blocking agents are known to cause cell membrane blebbing [213]. Thus, they were examined to replace physical processes to induce EV formation. Ingato et al. exposed a mouse lymphoma cell line to sulfhydryl-blocking agents, i.e., dithiothreitol (DTT) and paraformaldehyde (PFA), to prepare EVs [164]. Within 2 h, sulfhydryl blocking boosted EV production by more than tenfold compared with that by cells cultured in standard conditions for 48 h. EVs created using this approach had better cellular absorption and intracellular release of doxorubicin than liposomes. Using a mouse model, the authors showed that the doxorubicin-loaded, sulfhydryl-blocking-produced EVs were more effective in slowing down tumor growth than free doxorubicin and liposome-encapsulated doxorubicin. Doxorubicin, a chemotherapy drug, has also been identified to induce cancer cells to produce more EVs [19,146].

## 5. Other Factors Affecting Extracellular Vesicle Production

Apart from all the strategies mentioned above to boost the secretion of EVs, optimization of the cell culture parameters is critical for the large-scale production of EVs. Patel et al. found that small EV production was reduced when BM-MSCs were seeded at a high density of 10,000 cells/cm^2^ compared with the lower seeding density of 100 cells/cm^2^ [165]. The reduction in small EV particle concentration was very prominent at 50- to 105-fold at P2 to P5, based on the NTA data. The authors attributed the higher small EV production to higher small EV secretion due to the indirect cell–cell communication when cells are far apart in low-density culture. Direct cell–cell contact in high-density culture diminished the need for indirect cell–cell communication via EV secretion. The same observation of reduced EV production at higher cell seeding density was conserved for HDMECs, HEK cells, and human umbilical vein endothelial cells (HUVECs). In the same study, the authors found that the yield of small EVs increased when the medium collection frequency increased. Collecting EVs twice every 12 h (total 24 h), every 6 h (total 12 h), and every 3 h (total 6 h) increased the small EV yield by 1.6-fold, 2,4-fold, and 2.0-fold, respectively, compared with collecting EVs once at the later timepoints when the cells were seeded at a density of 100 cells/cm^2^. In terms of functionality, it was reported that the provascularization activity was reduced for small EVs collected from passage 5 cells. These findings suggested that prolonged cell expansion might diminish the therapeutic efficacy of the secreted EVs. The poorer therapeutic efficacy of the EVs secreted by the high-passage cells could be linked with cell senescence after long-term expansion. A reduction in EV secretion by cultured cells seeded at high seeding densities was also reported by Kim et al. [112], who found that BM-MSCs seeded in six-well plates at a density of 1.4 × 10^6^ cells/well produced fewer EVs than those seeded at a density of 1 × 10^5^ cells/well.

Several studies reported that stem cell differentiation affected the therapeutic potential of secreted EVs [214,215,216]. Interestingly, in one of these studies, the results showed that late osteogenic differentiated MSCs (day 21) secreted more small EVs than early osteogenic differentiated MSCs (day 3), while naïve MSCs secreted the lowest number of small EVs, even though the differences were not statistically significant [214]. The findings from this study indicated that cell differentiation influenced not only the quality but the quantity of the EVs secreted by stem cells.

EVs are among the channels utilized by senescent cells to remove harmful molecules from the cells (such as cytoplasmic DNA) to maintain cell homeostasis [166]. Senescent cells secrete more EVs, likely in response to the higher amounts of harmful molecules produced as cells age. Increased EV secretion by senescent cells was reported in both stem cells and cancer cells [166,167,168]. One of the aforementioned studies reported that small EVs secreted by senescent normal human diploid fibroblast (HDF) TIG-3 cells promoted the proliferation of human breast cancer MCF-7 cells, but small EVs secreted by presenescent TIG-3 cells did not [168]. The uptake of senescent EVs has been shown to induce cell senescence [217] and inhibit the osteogenic differentiation of MSCs [218]. Thus, senescent cells secrete more EVs, but the secreted EVs might not be usable in the clinic. The findings from these studies clearly showed that even though some techniques and culture conditions can increase EV production, the therapeutic potential of the produced EVs may be compromised. Thus, it is critical to examine the safety and efficacy of upscaled EVs.

## 6. Translation of EV-Based Therapeutics

Several hurdles are obstructing the clinical translation of EV-based therapeutics. These stumbling blocks include defining reliable and translatable cell sources, culture conditions, enrichment methods, characterization, and storage stability and determining the half-life and biodistribution of EVs [219,220,221]. In addition, donor variability, differences in the manufacturing process, and the use of xenogeneic reagents may profoundly impact the therapeutic activity of EVs [222]. Most studies have stored EVs in phosphate-buffered saline (PBS) at −80 °C for up to six months, but this may change the characteristics and functional performance of EVs [223]. Freeze-drying can prolong the shelf-life of EVs. However, the bioactivity of freeze-dried EVs also deteriorates with time. There is also a lack of information on the ideal route of administration and dosage regimen for EV therapies. Studies have revealed a short half-life of intravenously transfused exosomes of 2 to 15 min in circulation and accumulation of these EVs in the spleen, lungs, kidneys, and liver [224,225,226].

To comply with regulatory requirements, EV-based therapeutics must be produced in good manufacturing practice (GMP)-accredited facilities based on a standardized EV production protocol. Quality control is critical to ensure that EVs are consistently produced and to guarantee the quality standard of commercialized EV-based products produced in GMP facilities. Safety control must also be applied to secure the safety of EV donors and recipients. One crucial requirement for the release of EV-based therapeutics would be EV characterization [222]. The characterization of EVs, including protein marker and single-vesicle analysis, was discussed in the MISEV2018 guidelines [227]. Zhang et al. selected specific batches of small EVs with similar sizes and particles per µg protein to standardize the EVs used in their study [228]. Consistency of EV size, protein content, and particles per µg protein is crucial to attain reproducible results in clinical settings. However, the ideal characteristics of EVs depend on their functional activity and therapeutic use. EV protein, RNA, and lipid content and non-EV components should be added to the quality control list to identify the mechanism of action (MoA) of EV-based therapeutics, which is essential in clinical translation. However, the exact MoA of EV-based therapeutics is difficult to determine because of the heterogeneity of donor cells and secreted EVs. Furthermore, the selection of EV isolation methods is important, as different isolation methods may enrich EVs with different cargo, thus resulting in different efficacies and MoAs [229,230]. Last, the pathogen safety of EVs, especially on pathogenic viruses, must be emphasized, and the viral-removal steps should not affect the quality and functionality of EVs, especially for EVs produced by transgenic cells prepared via the viral-based transfection method. 

To date, there is still no gold-standard method for separating EVs from non-EV components. There exists a relative relationship between EV yield and purity [231,232]. Highly purifying EV separation methods always result in lower EV yield. On the other hand, although impurities may affect the functional activity of EVs, the protein coronae around EVs have been found to enhance the therapeutic efficacy of the EVs, especially for regenerative or immunomodulatory purposes [233,234,235]. Hence, the EV enrichment method must be chosen wisely according to the therapeutic use. Still, applying scalable, reproducible, and translatable EV isolation techniques is essential. Certain isolation methods can be used in combination to increase the efficiency of EV isolation, outperforming single-method isolation approaches. Watson et al. used a combination of GMP-compatible tangential flow filtration (TFF) and size-exclusion chromatography for large-scale EV production without compromising the EV functionality [236]. 

Provided that large-scale manufacturing is necessary for commercializing EV therapeutics, bioreactors appear to be the ideal platform, as they have been found to increase the yield and therapeutic efficacy of EVs. A recent study showed that bioreactors allowed consistent production of small EVs on a large scale [237]. In addition, disposable bioreactors are available for GMP-compliant cell cultures. Bioreactors enable more precise control of the culture environment (e.g., oxygen concentration, glucose concentration, pH, and temperature) to modulate EV secretion. However, more studies are needed to explore the effects of combined stimulation with multiple strategies on the yield, safety, and efficacy of the produced EVs. 

## 7. Conclusions

Ideally, EV preparation techniques should allow the production of large quantities of highly homogenous EVs in a short period. This is important to enhance the clinical translatability of EV therapeutics in the future and meet patient demands. The current review showed the emergence of various methods to achieve this, including 3D cultures, genetic manipulation, and physical, chemical, and physiological stimulation of EV secreting cells, as well as EV-mimetic nanovesicle preparation. These manipulations not only increase EV yield but alter EV cargo and functionality. Thus, careful evaluation of these techniques is vital to identify suitable large-scale EV production strategies that can increase yield without sacrificing efficacy or posing harmful risks. Moreover, it would be ideal if the production strategies also enhanced the therapeutic potential of the produced EVs. It is worth noting that quality control at all stages of development in line with GMP is needed for the successful translation of these approaches.

## Figures and Tables

**Figure 1 ijms-23-07986-f001:**
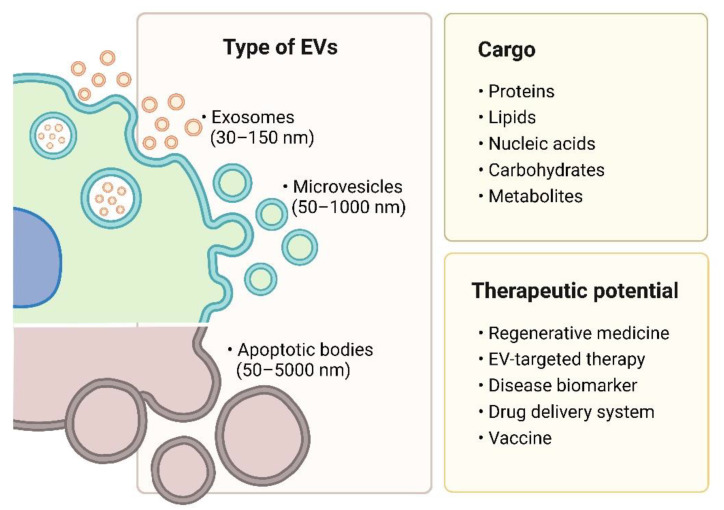
Type, cargo, and therapeutic potential of extracellular vesicles. Created with BioRender.com.

**Figure 2 ijms-23-07986-f002:**
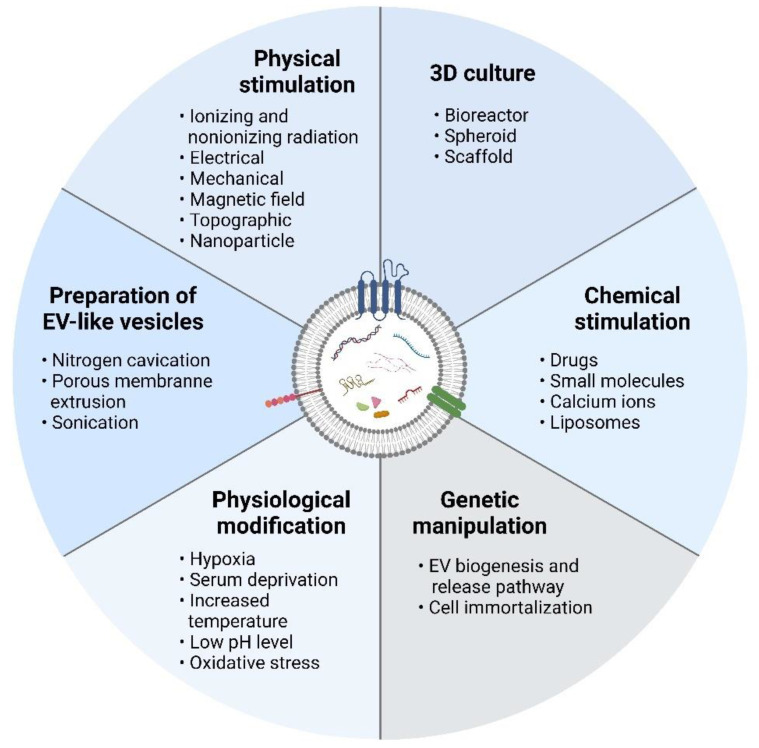
Scalable production of extracellular vesicles. Physical stimulation, chemical stimulation, 3D culture, physiological modification, and genetic manipulation can stimulate cells to produce more extracellular vesicles. In addition, EV-mimetic nanovesicles can be produced via nitrogen cavitation, porous membrane extrusion, and sonication. Created with BioRender.com.

**Table 1 ijms-23-07986-t001:** Strategies to increase EV production.

Strategies	Type of Induction	Cells	Exposure Period or Method	EV Isolation Method	Enhancement Factor	Therapeutic Application	References
3D culture	Hollow-fiber bioreactor (FiberCell Systems)	UC-MSCs	Cell expansion and medium conditioning	Ultracentrifugation	Bradford assay: 7.5-fold increase in small EV protein concentration	Possessed superior chondroprotective effects to those of 2D small EVs in vitro and in vivo	[105]
	Hollow-fiber bioreactor (FiberCell System)	UC-MSCs	Cell expansion and medium conditioning	Ultracentrifugation	BCA assay: 19.4-fold increase in small EV protein concentration	Possessed superior renoprotective efficacy to that of 2D small EVs in vitro and in vivo	[106]
	Vertical-Wheel™ bioreactors (VWBR)	BM-MSCs, AT-MSCs, and UC-MSCs	Cell expansion and medium conditioning	Precipitation (total exosome isolation reagent)	NTA: 4.0, 4.4, and 8.8-fold increases in small EV particle concentration for BM-MSCs, AT-MSCs, and UC-MSCs, respectively	Not reported	[107]
	Hollow-fiber bioreactor (FiberCell System)	hetIL-15-overexpressed HEK293 cells (clone 19.7)	Cell expansion and medium conditioning	Ultracentrifugation	Bradford assay: 40-fold increase in small EV protein concentration	Bioactivity of small EV-associated hetIL-15 was maintained (hetIL-15 activates NK cells)	[108]
	Hollow-fiber bioreactor (FiberCell System)	BM-MSCs	Cell expansion and medium conditioning	Precipitation (total exosome isolation reagent)	NTA: 1.9 × 10^10^ ± 1.1 × 10^10^ small EV articles/mL on day 1, 8.2 × 10^9^ ± 3.0 × 10^9^ small EV particles/mL on day 13, and 8.1 × 10^9^ ± 3.3 × 10^9^ small EV particles/mL on day 25	Possessed immunomodulatory properties	[109]
	Ultrafoam scaffolds (collagen type I)	MSCs	Cell expansion and medium conditioning	Precipitation (ExoQuick)	BCA assay: twofold increase in small EV protein concentration	Enhanced neurological functional recovery of traumatic brain injury model compared with 2D-culture and liposome groups	[110]
	3D-printed scaffold perfusion bioreactor	Human dermal microvascular endothelial cells (hDMECs)	Cell expansion and medium conditioning	Ultracentrifugation	NTA: 100- and 10,000-fold increases in small EV particle concentration on days 1 and 3, respectively; CD63 exoELISA: 14-fold increase in CD63+ EV concentration; BCA assay: 6.7-fold increase in small EV protein concentration, but decreased protein content per EV	Enhanced vascularization bioactivity in 3D-scaffold groups (bioreactor and static) pretreated with 100 nM ethanol	[111]
	3D spheroids	BM-MSCs	Cell expansion and medium conditioning	Precipitation (ExoQuick-TC)	Bradford assay: 2-fold increase in EV protein concentration for hanging-drop 3D spheroid culture; 2.4-fold increase for poly-HEMA coated-3D spheroid culture	Not reported	[112]
Physical stimulation	Ionizing radiation (X-ray: 2 Gy)	MCF7 breast epithelial cancer cells	4 h	Ultracentrifugation	TRPS: threefold increase in small EV particle concentration in the direct irradiated group; sixfold increase in bystander group	Identified that small EVs play a role in nontargeted effects of irradiation (cancer therapy)	[113]
	Gamma irradiation (1000 cGy) or hypoxia (1% O_2_)	Human lung cancer cell lines (LLC and A549)	12, 24, 36, or 48 h	Centrifugation	Flow cytometry: fourfold increase in EV particle concentration in both hypoxia and gamma irradiation treatment groups	Identified that the microenvironment caused EV change	[114]
	Ionizing radiation (X-ray: 4 Gy)	Human glioblastoma cell lines (LN18, U251, U87MG), glioblastoma stem-like cells (GBAM1 and GBMJ1), and astrocytes	12 to 48 h	Ultracentrifugation	NTA: 1.23- to 2.6-fold increases in small EV particle concentration	Increased cell migration and uptake efficiency, showing that intercellular signaling reacted to therapeutic radiation	[115]
	Ultraviolet irradiation stress treatment (UV; 40 W), low-pH culture medium treatment (LP; pH 4.0), high-temperature treatment (HT; 40 °C), H_2_O_2_ treatment (H_2_O_2_; 250 × 10−6 m), and hypoxic environment treatment (Hyp; 100% N_2_)	Human gastric cancer cells (MGC803) and human liver cancer cells (HepG2)	Not reported	Ultracentrifugation	BCA assay: 1.9-fold increases in small EV protein content in UV, LP, and HT treatments; 1.7-fold increase in H_2_O_2_; 1.5-fold increase in HYP	Increased uptake efficiency	[116]
	Photodynamic therapy (Foscan^®^ photosensitizer: 0.02, 0.08, 0.2, 0.5, 2 or 10 μM) and chemotherapeutic agent (doxorubicin: 0.1, 0.5, 2, 5, 10 and 50 μM)	Human prostatic cancer cells (PC-3)	2 h exposed to light for 5 s at a wavelength of 470 nm (7.5 J/cm^2^)	Not reported	NTA: 15- and 6-fold increases in large EV particle concentration for PDT and doxorubicin treatment, respectively	The released large EVs may counterbalance the desired regional limitation of a treatment and represent an underestimated source of adverse effects during PDT	[19]
	Acoustic irradiation: surface-reflected bulk waves (SRBWs, 4 W) and electromechanical hybrid surface (order of 10 MHz)	Human glioblastoma cells (U87-MG) and adenocarcinoma human alveolar basal epithelial cells (A549)	10 min followed by 30 min postexcitation incubation period	Column-based (PureExo^Ⓡ^ Exosome Isolation Kit)	AChE activity: 1.7-fold increase in small EV AChE activity in the first 30 min, followed by a reduction	Exosome therapy: cancer vaccine and biomarker	[117]
	Ionizing radiation (X-ray: 0 Gy; 0.1 Gy; 1 Gy; 10 Gy)	Neuroblastoma cell lines (SH-SY5Y and SK-N-BE)	3 h	Ultracentrifugation	Flow cytometry: 2.7-fold and 4.5-fold increases in small EV particle concentration with 0.1 Gy and 10 Gy radiation, respectively, for the SH-SY5Y cell line; 3.8-fold increase with 10 Gy radiation for the SK-N-BE cell line Spectrophotometric quantitation: 1.2-fold and 2.2-fold increases in small EV protein content with 0.1 Gy and 10 Gy radiation, respectively, for the SH-SY5Y cell line; 1.2-fold increase with 10 Gy radiation for the SK-N-BE cell line	Increased proliferation and invasiveness, showing side effects of radiation therapy	[18]
	Electrical stimulation (0.34 mA/cm^2^)	Melanoma cell line (B16F1) and murine fibroblast cell line (3T3)	1 h	Ultracentrifugation	TRPS: 1.26- and 1.7-fold increases in EV particle concentration for B16F1 and 3T3 cells, respectively	Not reported	[118]
	Cellular nanoporation (CNP)	Embryonic fibroblasts (MEFs) or bone marrow-derived dendritic cells (BMDCs)	4, 8, 12, 16, 20, and 24 h	Ultracentrifugation	DLS and NTA: 50-fold increase in EV particle concentration and >1000-fold increase in exosomal mRNA transcripts	Targeted therapy by transfer of desired peptides (through CD74) led to longer circulatory half-life, significantly inhibited glioma tumor growth in vivo, and prolonged survival	[119]
	Medium containing magnetic nanoparticles (Fe_3_O_4_: 50 µg/mL) and/or static magnetic field (SMF: 100 mT)	BMSCs	7 and 14 days	Ultracentrifugation	BCA assay: 1.4-fold increase in small EV protein concentration for Fe_3_O_4_ group; 1.7-fold increase for Fe_3_O_4_ + SMF group	Enhanced osteogenesis and angiogenesis in vitro and in vivo	[120]
	Cyclic stretch (20% elongation at a frequency of 10 cycles/min)	Periodontal ligament cells	24 h	Cell culture supernatant	CD63 ELISA Kit (PS Capture™ Exosome ELISA Kit): 33-fold increase in CD63+ EV concentration	Inhibited IL-1β production and pyroptosis of LPS-primed macrophage	[121]
	Micro-/nanotextured hierarchical titanium topography (native titanium specimens (SLM); SLM + 250 μm ZrO_2_ particles + 5% hydrofluoric acid (HF) (SLA); SLA + 5 M NaOH (SAH); SLA + 0.3 wt% ammonium fluoride (NH_4_F) + ethylene glycol (C_2_H_6_O_2_) solution (SAO))	BMSCs	During cell culture	Kit (EIQ3)	AChE activity: 1.1-fold, 1.7-fold, and 1.6-fold increases in small EV AChE activity in SLA, SAH, and SAO groups, respectively, compared with SLM group	Improved osseointegration in vitro and in vivo	[122]
	Platinum nanoparticles (10 µM)	Human lung epithelial adenocarcinoma cancer cells (A549)	24 h	Precipitation (ExoQuick)	BCA assay: 3.9-fold increase in small EV protein concentration; fluorescence polarization: 4.8-fold increase in small EV particle concentration; NTA: 4.1-fold increase in small EV particle concentration; EXOCET: 5.9-fold increase in small EV particle concentration	Not reported	[123]
	45S5 Bioglass^®^	Human MSCs	12 to 72 h or 48 h	Ultracentrifugation and ultrafiltration	AChE activity: No significant difference in small EV AChE concentration in the first 12 h; 1.3-, 1.4-, and 1.6-fold increases at 24, 48, and 72 h, respectively NTA: No significant differences in small EV particle concentration in the first 12 h; 2.4-, 1.8-, and 2.0-fold increases at 24, 48, and 72 h, respectively EXOCET kit: 2.1-fold increase in small EV particle concentration at 48 h HSFCM: 5.4-fold increase in small EV particle concentration at 48 h	Promoted vascularization of umbilical vein endothelial cells in vitro and in vivo	[124]
Chemical stimulation	Sodium iodoacetate and 2,4-dinitrophenol (IAA/DNP) (in vitro: 1 or 10 µM; ex vivo: 5, 10, or 30 µM; in vivo: 0.195 or 0.975 μmol)	UMSCC47, PCI-13, Mel526, SVEC4–10 (in vitro); murine kidney tissue explant (ex vivo); mice (in vivo)	72 h (in vitro); 48 h (ex vivo); 14 days (in vivo)	Size-exclusion chromatography	BCA assay: 3- to 16-; 1.8-, and 2.9-fold increases in small EV protein concentration in vitro, ex vivo, and in vivo, respectively	Possessed similar biological properties and functional effects on endothelial cells (SVEC4-10)	[125]
	Fenoterol, norepinephrine, N-methyldopamine, mephenesin, and forskolin	BMSCs	24 h	Ultracentrifugation	NTA: 1.7- to 2.3-fold increase in small EV particle concentration, which further increased when combining compounds (2.5- to 3-fold)	Possessed regenerative activities as control small EVs	[126]
	Suxiao Jiuxin pill, tetramethylpyrazine, or borneol	Murine cardiac MSCs	48 h	Precipitation (polyethylene glycol 4000)	AChE activity: 3.4-fold, 2.4-fold, and 1.3-fold increases in small EV AChE activity in Suxiao Jiuxin pill, tetramethylpyrazine, and borneol treatments, respectively	Not reported	[127]
	Adiponectin (20 μg/mL) from serum collected from APN-knockout mice	T-cadherin-expressing murine vascular endothelial cells (F2T cells)	36 h	Ultracentrifugation	AChE activity: 7.8-fold increase in small EV AChE activity; NTA: 2.9-fold increase in small EV particle concentration	Adiponectin-induced small EV release affected ceramide metabolism, which could be helpful for adiponectin-related organ protection therapy	[128]
	High-molecular-weight adiponectin (20 μg/mL; in vitro) or pioglitazone (30 mg/kg twice a day; in vivo)	Human adipose tissue-derived MSCs	48 h (in vitro) and two weeks (in vivo)	Ultracentrifugation	Densitometry of Western blot: increased small EV production in vitro and in vivo; NTA: 3.3-fold increase in small EV particle concentration in vitro	Augmented the cardioprotective effects of MSCs in transverse aortic constriction-operated mice	[129]
	Docosahexaenoic acid (DHA; 100 μM)	Human breast cancer cells (MCF7 and MDA-MB-231)	24 h	Ultracentrifugation or precipitation (ExoQuick-TC reagent)	CD63-GFP fluorescent spectrometry: 1.1-fold increase in CD63+ EV concentration in MCF7 and MDA-MB-231 cell lines	Increased RNA content in breast cancer CD63+ EVs promoted anticancer and anti-angiogenic activity	[130]
	Sodium ionophore (Monensin; 1, 5, 10 µM), calcium ionophore (A23187; 1 µM), or human transferrin (20 µg/mL)	Human erythroleukemia cell line (K562)	7 h for monensin and A23187; 12 h for transferrin treatment	Ultracentrifugation	AChE activity: 20%, 71.5%, and 97.6% increases in EV AChE activity with 1, 5, and 10 µM of monensin; 1.7-fold increase with A23187; 1.4-fold with transferrin	Not reported	[131]
	Sodium ionophore (Monensin: 7 µM)	Rab11-transfected human erythroleukemia cell line (K562)	7 h	Ultracentrifugation	AChE activity: 2.0-, 1.8-, 3.8-fold, and 3.7-fold increases in EV AChE activity with monensin treatment in vector, Rab11 wildtype, Rab11 Q70L (a GTPase-deficient mutant), and Rab11 S25N (aGTP-binding deficient mutant) cells, respectively	Not reported	[132]
	Calcium phosphate (CaP) particles (500 and 1000 μg/mL)	Macrophage-like cells (RAW264.7) and monocyte-like cells (THP-1)	1, 2, 4, 6, 24, 48, and 72 h	Precipitation (total exosome isolation kit)	EXOCET exosome quantitation assay kit: 2-and 2.5-fold increases in small EV particle concentration at 72 h with 500 μg/mL CaP for RAW264.7 and THP-1 cell lines, respectively	Not reported	[133]
	Ionomycin (2.5 µM) and TGFβ-1 (5 ng/mL)	Human breast carcinoma cell line (MDA-MB-231), human lung carcinoma line (A549), and human pancreatic carcinoma line (Panc-1)	30 min of ionomycin treatment 24 h of TGFβ-1 treatment	Density gradient ultracentrifugation	CD63^+^ Slot blot: 5- and 3-fold increases in CD63+ EV concentration for ionomycin and TGFβ-1 treatments, respectively	Not reported	[134]
	Phosphorothioate (PS) B-class CpG oligonucleotides (ODN 2006PS; 2 μM), *S. salar* DNA (15 μg/mL), or *E. coli* DNA (15 μg/mL)	Salmon head kidney leukocytes (HKLs), Atlantic salmon kidney cells (ASK cells), chinook salmon embryo cells (CHSE-214 cells), or HEK293T cells	1 h (2006PS, *S. salar* or *E. coli* DNA) for HEK293T cells	Ultracentrifugation	Densitometry of Western blot (Alix): 10.1-, 16.7-, and 9.1-fold increases in Alix+ EV protein content for ODN 2006PS-treated HKLs, ASK cells, and CHSE-214 cells, respectively; for HEK293T cells, 3.3-, 2.1-, and 9-fold increases in Alix+ EV protein content for 2006PS, *S. salar* DNA, and *E. coli* DNA groups, respectively	Not reported	[135]
	Cationic bare liposomes (CL: HSPC-based or DOPE-based; 0.5 to 2 mM) or neutral bare liposomes (NL; 0.5 to 2 mM)	Murine colorectal cancer cell line (C26), murine melanoma cell line (B16BL6), human gastric cancer cell line (MKN45), and human colorectal cancer cell line (DLD-1)	48 h	Ultracentrifugation or precipitation (ExoQuick-TC^TM^)	Bio-Rad DC^®^ protein assay: 2.5-, 2.3-, 1.7-, and 1.8-fold increases in EV protein concentration with 2 mM of NL for C26, B16B16, MKN45, and DLD-1 cell lines, respectively; 3.4-, 3.4-, 3.7-, and 2.9-fold increases in EV protein concentration with 2 mM of HSPC-based CL for C26, B16B16, MKN45, and DLD-1 cell lines, respectively. DOPE-based CLs further increased EV protein concentration (up to 3.17-fold)	Liposome-stimulated EVs showed higher cellular uptake	[136]
Physiological modification	Hypoxic (1% O_2_, 0.1% O_2_, or 1 mM DMOG)	Breast cancer cell lines MCF7, SKBR3, and MDAMB 231	24 h (1% O2) or 48 h (0.1% O2 and DMOG)	Precipitation (ExoQuick^TM^)	NTA: Up to 1.41-fold, 1.94-fold, and 1.45-fold increases in small EV particle concentration at 1% O_2_, 0.1% O_2_, and 1 mM DMOG, respectively	Highlighted the importance of the study of EV-mediated pathological hypoxic signaling in tumor progression	[137]
	Hypoxic (5% O_2_)	Human umbilical cord MSCs	24 h	Ultracentrifugation	NTA: twofold increase in small EV particle concentration	Better attenuated OVA-induced chronic airway inflammation and lung parenchyma fibrosis in mice	[138]
	Hypoxic (1% O_2_)	BMSCs	48 h	Ultrafiltration and density gradient ultracentrifugation	BCA assay: 1.4-fold increase in small EV protein concentration	Better protected cartilage from degeneration and slowed down the progression of OA in vitro and in vivo	[139]
	Hypoxic (1% O_2_)	BMSCs	48 h	Ultrafiltration and density gradient ultracentrifugation	BCA assay: 1.4-fold increase in small EV protein concentration	Promoted to a greater extent functional behavioral recovery in mice and M1 to M2 phenotype polarization in vivo and in vitro	[140]
	Hypoxic (1% O_2_) and/or serum-free stimulation	UC-MSCs	72 h	Ultracentrifugation	Bradford assay: 5.6-, 4.3-, and 7.5-fold increases in CD29+, CD44+, CD73+, CD31−, and CD45− EVs (authors identified the EVs as microvesicles) under hypoxic conditions, serum-free stimulation, and hypoxic and serum-free conditions, respectively	CD29+, CD44+, CD73+, CD31−, and CD45− EVs promoted angiogenesis and were superior in hypoxia endothelial cells	[141]
	Hypoxic (0.5% O_2_)	Mouse MSCs	24 h	Ultracentrifugation	NTA: 1.3-fold increase in small EV particle concentration	Superior ability in proangiogenesis and antiapoptosis in vitro and cardiac protection in vivo	[142]
	Lentivirus pWPI-HIF-1α-GFP transduction	Dental pulp MSCs	Lentivirus transduction	Ultracentrifugation	Densitometry of Western blot: 3.3-, 6.3-, and 1.5-fold increases in CD63, CD9, and CD81 density AChE activity; 2.1-fold increase in small EV AChE activity	Superior angiogenic ability in vitro and in vivo via enhanced expression of the Notch ligand Jagged1	[143]
	Peripheral arterial disease conditions (0% serum and 1% O_2_)	BMSCs	40 h	Centrifugation (higher density EVs (claimed as microvesicles)) or ultracentrifugation (lower density EVs (claimed as exosomes))	BCA assay: 9-fold decrease in high-density EV protein concentration; 6.6-fold increase in low-density EV protein concentration	EVs contained a robust profile of angiogenic signaling proteins and induced angiogenesis	[144]
	Heat stress (42 °C)	Epstein–Barr virus (EBV)-immortalized human B-lymphoblastoid cell lines (B-LCL) and Jurkat cell line	3 h	Density gradient ultracentrifugation	BCA assay: 1.25-fold increase in EV protein concentration for all cell lines	Significantly increased heat shock proteins of cells and EVs but did not trigger dendritic cell maturation	[145]
	Heat stress (40 °C), oxidative stress, cells were treated for 2 h with 100 μM and 50 μM H_2_O_2_	Jurkat and Raji cells	1 h	Density gradient ultracentrifugation	Densitometry of Western blot (CD63): 3- and 15-fold increases in CD63+ EV protein concentration after thermal and oxidative stress, respectively, for Jurkat cells; 22- and 32-fold increases after thermal and oxidative stress for Raji cells	Partly provided a mechanistic explanation of the clinically observed NK-cell dysfunction in patients suffering from leukemia/lymphoma, which could be further impaired in conditions of cellular stress	[21]
	Cytostatic stress (0.6 μM doxorubicin), heat stress (42 °C), oxidative stress (2.5 μg/mL light-induced Ag-TiO_2_)	B16F1 mouse melanoma cell	72 h heat stress (3 × 2 h)	Ultracentrifugation	NTA: 3.6-, 2.5-, and 1.6-fold increases in small EV particle concentration after doxorubicin, heat stress, and oxidative stress, respectively	Microenvironmental conditions altered small EV cargoes and explained the importance of determining therapy-induced host response	[146]
	pH (pH 6.0)	Mel1 melanoma cell lines	Cell culture and medium conditioning	Density gradient ultracentrifugation	BCA assay: 3.2- and 6.3-fold increases in Lamp-2+, CD81+, and Rab 5B+ EVs on days 3 and 4, respectively	Acidic microenvironment favored EV-to-cell fusion	[12]
	pH (pH 4, 7, and 11)	HEK 293	30 min	Precipitation (ExoQuick isolation kit)	BCA assay: 5-fold increase in CD9+, CD63+, and Hsp70+ EV protein concentration at pH 4, while pH 11 gave a negative result (3.5-fold decrease)	Acidic pH could increase the stability of EVs in vitro	[147]
Genetic manipulation	PLD2 cDNA electroporation	RBL-2H3	Murine PLD2 cDNA electroporation	Ionomycin degranulation then ultracentrifugation	FACS: Twofold increase in BODIPY-ceramide labeled EV concentration	Not reported	[148]
	Rab13 knockdown	Mutant KRAS DKO-1 colorectal cancer cells	Rab13 shRNA transfection	Ultracentrifugation	NTA: 14.3-fold decrease in small EV production when Rab13 was knocked down in mutant KRAS cells	Not reported	[149]
	Rab27a or Rab27b knockdown	HeLa cell line	Transfection	Ultracentrifugation	Bradford assay: 2.5- and 2.2-fold decreases in small EV protein content for Rab27a knockdown and Rab27b knockdown cells, respectively Densitometry of Western blot (HLA-DR, Hsc70, Tsg101): 19.5-, 5.8-, and 3.1-fold decreases in HLA-DR, Hsc70, and Tsg101 protein density of small EVs, respectively, for Rab27a knockdown cells; 2.5-, 2.4-, and 1.6-fold decreases in HLA-DR, Hsc70, and Tsg101 protein density, respectively, for Rab27b knockdown cells	Not reported	[150]
	CD9 lentiviral transduction	HEK293, HEK293FT, Raji, Jurkat, and HeLa	pLenti6.3-CD9_GFP_ lentiviral transduction	PEG centrifugation	NTA: 2.5-fold increase in small EV particle concentration in HEK293-CD9_GFP_ cells; more than 3-fold increase in HEK293FT-CD9_GFP_; 2-fold increase in Raji-CD9_GFP_ and Jurkat-CD9_GFP_; no significant difference for HeLa-CD9_GFP_	Not reported	[151]
	Liver kinase B1 (LKB1) lentiviral transduction	H460	pCDH-LKB1 lentiviral transduction	Ultracentrifugation	NTA: 12.5-fold increase in small EV particle concentration	Enhanced cell migration ability and showed the ambiguity of cancer therapy targeting LKB1 function	[152]
	Upregulation of EIF3C	Hepatocellular carcinoma cell lines: PLC5 and SNU-449	NA	Precipitation (total exosome isolation reagent)	Electron microscopy and NTA: Increased small EV particle concentration	EIF3C may act as a cancer target in cancer therapy	[153]
	PIKfyve inhibition (apilimod or siRNA transfection)	Human prostate cancer epithelial cell line (PC-3)	Apilimod or siRNA transfection	Ultracentrifugation and OptiPrep density gradient centrifugation	NTA and BCA assay: Apilimod treatment increased small EV particle concentration 1.6-fold and small EV protein content 1.4-fold; siRNA transfection increased small EV particle concentration 1.5-fold	Not reported	[154]
	MYC gene overexpression	hESC-MSC (HuES9.E1 MSCs)	GFP- or MYC-containing lentivirus transduction	HPLC	Not reported	Provided an infinite supply of cells for the production of small EVs with cardioprotective activity	[155]
Preparation of EV-mimetic nanovesicles	Nitrogen cavitation (400–500 psi at 0 °C)	Promyelocytic leukemia cell line (HL-60)	20 min	Centrifugation	BCA assay: 16-fold increase in EV-mimetic nanovesicle protein concentration	Prevented sepsis-induced inflammation and increased animal survival after loading of piceatannol	[156]
	Sonication (42 kHz and a power of 100 W) of the mixture of macrophage membranes and nanoparticle cores	Murine J774 macrophage cell line	2 min	Macrophage membranes mixed with nanoparticle cores at the ratio of 1:1	Not reported	Promoted proinflammatory cytokine sequestration and endotoxin neutralization in vitro and in vivo	[157]
	Ultrasonication (20%)	Human umbilical cord-MSCs	1 min	Centrifugation	18.5-fold increase in EV-mimetic nanovesicle production	Enhanced skin rejuvenation and promoted wound healing in vitro and in vivo	[158]
	Serial extrusion through filters with diminishing pore sizes	Human U937 monocytic cells	Not reported	OptiPrep density gradient ultracentrifugation	NTA: 100-fold increase in doxorubicin-loaded EV-mimetic nanovesicle particle concentration	Targeted delivery of chemotherapeutic drug had a similar antitumor effect to that of doxorubicin-loaded natural EVs in vitro and in vivo	[159]
	Serial extrusion through filters with diminishing pore sizes	Mouse embryonic fibroblasts NIH3T3 and human U937 monocytic cells	Not reported	Two-step OptiPrep density gradient ultracentrifugation	Not reported	Therapeutic vesicles (c-Myc siRNA-loaded nanovesicles) targeted diseases associated with c-Myc overexpression	[160]
	Serial extrusion through filters with diminishing pore sizes	Adipose-derived stem cells	Not reported	Two-step OptiPrep density gradient ultracentrifugation	NTA: 30-fold increase in EV-mimetic nanovesicle particle concentration	Possessed similar regenerative effects to those of natural EVs in vitro and enhanced regenerative ability in emphysema mouse model	[161]
	Serial extrusion through filters with diminishing pore sizes		Not reported	Two-step OptiPrep density gradient ultracentrifugation	BCA assay: 100-fold increase in EV-mimetic nanovesicle protein concentration	Promoted hepatocyte proliferation in vitro and liver regeneration in vivo	[162]
	Alkaline solutions (sodium carbonate solution)	Human U937 monocytes	Not reported	Sonication then density gradient ultracentrifugation	Bradford assay: 200-fold increase in EV-mimetic nanovesicle protein concentration	Reduced the release of IL-8 from OMV-treated endothelial cells in vitro and mitigated the symptoms of OMV-induced SIRS in vivo by dexamethasone-loaded EV-mimetic nanovesicles	[163]
	Sulfhydryl-blocking agents (dithiothreitol (DTT; 2 mM) and paraformaldehyde (PFA; 25 mM))	Mouse lymphoma cell line (EL4)	2 h	Ultracentrifugation and centrifugal filtration	BCA assay: More than 10-fold increase in EV-mimetic nanovesicle protein concentration	Better cellular absorption and intracellular release of doxorubicin than liposomes; more effective in slowing down tumor growth than free doxorubicin and liposome-encapsulated doxorubicin	[164]
Other factors	Seeding density (1 × 10^2^ cells/cm^2^ or 1 × 10^4^ cells/cm^2^)	BMSCs	Not reported	Ultracentrifugation and ultrafiltration	NTA: 100-, 85-, 105-, and 50-fold increases in small EV particle concentration with 1 × 10^2^ cells/cm^2^ seeding density compared with 1 × 10^4^ cells/cm^2^ seeding density at passages 2, 3, 4, and 5, respectively CD63 ELISA: 126-, 152-, 201-, and 126-fold increases in CD63+ EVs with 1 × 10^2^ cells/cm^2^ seeding density	Not reported	[165]
	Collection frequency (single or double collection)	BMSCs	12 and/or 24 h collection time(s)	Ultracentrifugation and ultrafiltration	NTA: 1.6- to 2.6-fold increase in small EV particle concentration with two EV collections compared with a single collection in 24 h	Not reported	[165]
	Cellular senescence (presenescent and senescent cells)	Primary normal human diploid fibroblasts (TIG-3 cells)	Serial passage or ectopic expression of oncogenic Ras	Density gradient ultracentrifugation	NTA: Significant increase in small EV particle concentration in senescent cells	Revealed the role of small EV secretion in the maintenance of cellular homeostasis	[166]
	Cellular senescence (presenescent and senescent cells)	Human prostate cancer cells (22Rv1) and human dermal fibroblasts (NHDFs)	Replicative senescence (serial passage) or accelerated senescence (irradiation: 4 Gy)	Ultracentrifugation	Vybrant DiI labeling: 3- and 15-fold increases in Vybrant Dil-labeled EV particles in accelerated senescent and replicative senescent cells, respectively	Not reported	[167]
	Cellular senescence (presenescent and senescent cells)	Human diploid fibroblasts (HDFs; TIG-3 cells)	Replicative senescence (serial passage) or induced senescence (oncogenic Ras or doxorubicin induction)	Ultracentrifugation	NTA: 46.4-, 16.7-, and 6.8-fold increases in small EV particle concentration for replicative senescent cells, oncogenic Ras-induced senescent cells, and doxorubicin-induced senescent cells, respectively	Small EVs from senescent cells promoted the proliferation of human breast cancer MCF-7 cells, showing the protumorigenic effect of senescent cells	[168]

## Data Availability

Not applicable.

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
