# Peer review of "Scalable Production of Extracellular Vesicles and Its Therapeutic Values: A Review"

_ijms, 2022, doi:10.3390/ijms23147986_

Round 1

Reviewer 1 Report

n/a

Author Response

There are no comments or suggestions from the reviewer.

Reviewer 2 Report

Scalable Production of Extracellular Vesicles and Its Therapeutic Values: A Review

The ,manuscript has been improved and is of interest to people interested in the subject.

However,  the next should be clarified.

Please mention how is that exosomes had higher affinity to cancer cells and for other tissues.

“Cancer cell-derived EVs have been reported to have a higher affinity to cancer cells”.

Page 1 line 49- 51

Please explain why exosome possess a low risk of tumorogenicity  due to nanosize

Stem cell-derived EVs carry a low risk of tumorigenicity and allogeneic immune rejection and minimal risk of microvascular occlusion during intravascular administration because of their nano-size [4,5].

Author Response

Point 1: Please mention how is that exosomes had a higher affinity to cancer cells and for other tissues.

“Cancer cell-derived EVs have been reported to have a higher affinity to cancer cells”.

Response 1: Thank you for the comment. The higher affinity of EVs to cancer cells has been further elaborated: “Cancer cell-derived EVs have a higher affinity to cancer cells due to the unique protein and lipid composition that facilitates binding or internalization of EVs in cancer cells [10–12].” (page 2, line 57–59)

Point 2: Please explain why exosomes possess a low risk of tumorigenicity due to nanosize

Stem cell-derived EVs carry a low risk of tumorigenicity and allogeneic immune rejection and minimal risk of microvascular occlusion during intravascular administration because of their nano-size [4,5].

Response 2: Thank you for the comment to point out the confusion. The “nano-size” refers to the minimal risk of microvascular occlusion during intravascular administration, not for tumorigenicity and allogeneic immune rejection. The sentence has been rephrased as “Stem cell-derived EVs carry a low risk of tumorigenicity and allogeneic immune rejection as well as minimal risk of microvascular occlusion during intravascular administration because of their nano-size [4,5]” to avoid the confusion.

Author Response

There are no comments or suggestions from the reviewer.

This manuscript is a resubmission of an earlier submission. The following is a list of the peer review reports and author responses from that submission.

Round 1

Reviewer 1 Report

Scalable Production of Extracellular Vesicles: A Review

The manuscript is well written and organized. However, the authors need to address the following comments in the revision to improve the understanding of the study and strengthen it.

Is necessary to explain the difficulties to exploit and purify at the GMP level the different sizes of nanoparticles or vesicles. Mentioned some strategies of purification that can be scalable are indispensable.

For commercialization of exosome or vesicle, explain what the size is ideal, and the analysis required to identify and select it. And its properties

Explain the difference in a table or figure about the use of an extract whole obtained under the same methodology mentioned in figure 2 and only exosomes or vesicles.

Please mention the estimated time life of different exosomes and their activity and conditions of storage.

Reviewer 2 Report

The authors reviewed how to increase the EV yield. THe reviewer has the following comments:

  • The authors mentioned stem cells as a start in the introduction. Consider adding stem cells in the title. Scale-up of EV is only a small section of the review. The authors may consider changing to a different title. 
  • Scale-up for all three EV subtypes? OR only for exosomes? the authors need to clarify this. Apoptotic bodies are not necessarily 'good ones'.
  •  Figure 1: please double check the EV size. MV can be as small as 50nm. Exosomes can be as big as 200nm, Apoptotic bodies can be as small as 50nm.
  • 3. Therapeutic application of EVs. The authors should clarify which type of EVs are reviewed in this section. consider adding saliva and oral biofluid for periodontal disease papers here. 
  • The authors should list all studies mentioned in Section 4 Strategies to Increase Production of Extracellular Vesicles in a table for readers to understand. For 3D culture, how about 3D scaffolds? they are missing. Highlight the in vivo studies for scaled EV. 

Reviewer 3 Report

This review provides a comprehensive overview of the types and therapeutic values of extracellular vesicles in medical science. In addition, the manuscript summarizes strategies to increase production of extracellular vesicles, such as bioreactor, mechanical stimulation, electrical stimulation, thermal stimulation, magnetic field stimulation, topographic clue, hypoxia, serum-deprivation, pH modification, exposure to small molecules, exposure to nanoparticles, increasing the intracellular calcium concentration, and genetic modification. The abstract in the manuscript puts forward the purpose, significance and function of the review. The author compares the views and theoretical basis of different researches on the compatriots of extracellular vesicles, and further clarifies the research status of extracellular vesicles and the author's own opinions. Overall, this review provides a broad understanding of extracellular vesicles and helps advance research in this area. But the manuscript still needs improvement.

  1. 5 appears on page 13 and page 14 of the manuscript, which may be a writing error. Please correct it. 
  2. Some content should be discussed in the review. It is necessary to compare the experimental results in the current researches, mention the potential application value of extracellular vesicles, point out the limitations of the current study, and indicate the direction and possibility of follow-up research.
